# Association of Metabolites, Nutrients, and Toxins in Maternal and Cord Serum with Asthma, IgE, SPT, FeNO, and Lung Function in Offspring

**DOI:** 10.3390/metabo13060737

**Published:** 2023-06-09

**Authors:** Wilfried Karmaus, Parnian Kheirkhah Rahimabad, Ngan Pham, Nandini Mukherjee, Su Chen, Thilani M. Anthony, Hasan S. Arshad, Aniruddha Rathod, Nahid Sultana, A. Daniel Jones

**Affiliations:** 1Division of Epidemiology, Biostatistics, and Environmental Health, School of Public Health, University of Memphis, Memphis, TN 38152, USA; parnian.k@memphis.edu (P.K.R.); nahidkmc@gmail.com (N.S.); nganphamphuong@gmail.com (N.P.); 2Department of Epidemiology, Fay W. Boozman College of Public Health, University of Arkansas for Medical Sciences, Little Rock, AR 72205, USA; nmukherjee@uams.edu; 3Department of Biostatistics, College of Public Health, University of Nebraska Medical Center, Omaha, NE 68198-4375, USA; suchen@unmc.edu; 4Department of Biochemistry & Molecular Biology, Michigan State University, East Lansing, MI 48824, USA; thilani@chemistry.msu.edu (T.M.A.); jonesar4@msu.edu (A.D.J.); 5Clinical and Experimental Sciences, Faculty of Medicine, University of Southampton, Southampton SO17 1BJ, UK; s.h.arshad@soton.ac.uk; 6David Hide Asthma and Allergy Research Centre, Isle of Wight PO30 5TG, UK; 7Peter O’Donnell Jr. School of Public Health, University of Texas Southwestern Medical Center, Dallas, TX 75390, USA; aniruddhabhadresh.rathod@utsouthwestern.edu

**Keywords:** metabolites, nutrients, toxins, maternal serum, cord serum, asthma, allergy, lung function, phthalates, triacylglycerol, hypoxanthine, polyphenol syringic acid

## Abstract

The role of metabolites, nutrients, and toxins (MNTs) in sera at the end of pregnancy and of their association with offspring respiratory and allergic disorders is underexplored. Untargeted approaches detecting a variety of compounds, known and unknown, are limited. In this cohort study, we first aimed at discovering associations of MNTs in grandmaternal (F0) serum with asthma, immunoglobulin E, skin prick tests, exhaled nitric oxide, and lung function parameters in their parental (F1) offspring. Second, for replication, we tested the identified associations of MNTs with disorders in their grandchildren (F2-offspring) based on F2 cord serum. The statistical analyses were sex-stratified. Using liquid chromatography/high-resolution mass spectrometry in F0, we detected signals for 2286 negative-ion lipids, 59 positive-ion lipids, and 6331 polar MNTs. Nine MNTs (one unknown MNT) discovered in F0-F1 and replicated in F2 showed higher risks of respiratory/allergic outcomes. Twelve MNTs (four unknowns) constituted a potential protection in F1 and F2. We recognized MNTs not yet considered candidates for respiratory/allergic outcomes: a phthalate plasticizer, an antihistamine, a bile acid metabolite, tryptophan metabolites, a hemiterpenoid glycoside, triacylglycerols, hypoxanthine, and polyphenol syringic acid. The findings suggest that MNTs are aspirants for clinical trials to prevent adverse respiratory/allergic outcomes.

## 1. Introduction

The developmental origins of health and disease concept postulates that metabolic programming influenced by the early life environment alters the risk of disease later in life [1]. Metabolites, nutrients, and toxins (MNTs) encompass a wide range of endogenous and exogenous biochemicals including carbohydrates, amino acids, organic acids, nucleotides, lipids, steroids, vitamins, products of exposure to smoking, medications, and environmental xenobiotics, and other substances contributed by diet, microbiomes, and interactions between them and host metabolism. MNTs can serve as biomarkers of exposures and various metabolic pathways [2]. Investigating metabolites associated with respiratory and allergic diseases offers the opportunity to discover risks and protective factors as well as suggestive mechanisms of disease pathogenesis that may serve as markers of such chronic conditions developing later in life [3,4,5].

Among respiratory and allergic diseases, asthma is a common chronic condition in children and its development involves a complex interface of genetics and epigenetic regulations [6]. It is characterized by airway hyperresponsiveness and airway inflammation [7]. Allergic sensitization plays a major role in the development of asthma [8] and it is assessed by positive skin prick testing (SPT) [9]. Total IgE indicates an overall risk for allergic disease [10]. Several studies have also suggested the usefulness of Fractionated exhaled Nitric Oxide (FeNO) measurements for assessment of asthma as its expression increases with airway inflammation [11]. In addition, lung function markers such as forced vital capacity (FVC), forced exhalation volume in one second (FEV1), their ratio FEV1/FVC, and forced expiratory flow between 25% and 75% of FVC (FEF_25–75%_) provide reliable assessments of the pulmonary function.

Recent studies that have investigated metabolic signatures of asthma often focused on a targeted measurement approach for a limited number of known endogenous metabolites in children and adults with asthma, often measuring metabolites in urine or breath and, less frequently, in serum [12]. Some metabolites have been identified as potential regulators and markers of immune responses leading to allergy [13,14]. In addition, metabolic signatures of pulmonary function have been identified [15,16]. One recent and rare investigation of associations of maternal metabolomes in pregnancy with asthma in offspring in a U.S. cohort found that plasma levels of several metabolites attributed to coffee consumption were protective against asthma in offspring [17]. However, our understanding of the range of MNTs in human sera is immature, particularly for those contributed by exogenous exposures and interactions with microbiomes, and the role of maternal MNTs during pregnancy on inflammation markers/conditions of their offspring’s respiratory and allergic development has been underexplored.

In this study, we first aimed at discovering associations of MNTs in grandmaternal (F0) serum collected before birth with asthma, IgE, SPT, FeNO, and lung function parameters in their F1 offspring (parents of the F2 generation). To test replication of the findings in F0-F1, we associated F2 cord serum MNTs with disorders in F2 offspring. Since gender differences in the natural history of asthma [18], allergic sensitization [19], and lung function parameters have become obvious [20,21], the analytical approach was sex-stratified. Compared to targeted approaches, systematically analyzing an untargeted source of MNTs provides a unique opportunity to identify novel associations and compare the importance of different MNTs.

## 2. Methods

### 2.1. Study Population

The Isle of Wight birth cohort (IOWBC) is a population-based cohort established on the Isle of Wight, UK, to prospectively study the natural history of allergic diseases among children. The birth cohort consists of children born between 1 January 1989, and 28 February 1990. Of 1536 pregnancies, parents during this period were contacted, and subsequently, after exclusion due to pregnancy failure and missing written consent, 1456 infants were enrolled. Follow-ups were conducted through detailed interviews and examinations for each child at ages 1, 2, 4, 10, 18, and 26 years. The IOWBC includes three generations: F0 parents of the original cohort, F1 original cohort members, and F2 the offspring of F1. In this study, we focused on the F1 generation and their F0 mothers (data collected in 1989–1990) and the F2 generation (data collected in 2010–2019) and their F1 mothers or female spouses of F1 fathers. Details of the IOWBC have been described elsewhere [22,23].

### 2.2. Exposures in the F0 and F2 Generations

Grandmaternal blood samples from F0 participants and cord blood samples of the F2 generation were collected before and at birth, respectively. F0- and F2 serum were aliquoted and fractionated into organic and aqueous phases to measure polar MNTs in the aqueous phase, and in the organic phase, lipids detected as negative ions, and neutral lipids including triacylglycerols and cholesterol esters detected as positive ions.

### 2.3. Sample Preparation and Processing

Serum specimens were grouped, processed, and analyzed in random order. Each batch included analyses of multiple blanks, pooled quality control extracts, and extracts of reference serum. Since volumes were limited, aliquots of 20 μL were extracted from sera using a modified Matyash two-phase protocol into water-soluble and organic-soluble fractions with each extraction tube containing 25 pmol cotinine-*d*_3_ as internal standard plus additional secondary stable isotope-labeled internal standards. The polar (lower) fraction was evaporated to dryness under vacuum using a SpeedVac without heat application, and the non-polar (upper) layer was evaporated to dryness using a nitrogen evaporator. Non-polar fractions were dissolved in 1 mL of 2-propanol/water (90:10 *v*/*v*) and polar fractions in 200 µL of acetonitrile/water (90:10 *v*/*v*), and aliquots were transferred to glass autosampler vials.

### 2.4. Profiling of MNTs Using Liquid Chromatography/High Resolution Mass Spectrometry (LC/HRMS)

Profiling of polar fraction metabolites was executed using a QExactive mass spectrometer (Thermo Electron North America LLC, Madison, WI, USA) interfaced to a Thermo Vanquish Flex binary pump and auto-sampler equipped with an Acquity BEH Amide column (10 cm × 1.0 mm, 1.7 μm, Waters, Milford, MA, USA) for HILIC chromatographic separation, with analysis performed using positive-ion mode electrospray ionization (ESI) using full scan/all-ions fragmentation. Organic-soluble (non-polar) fractions were analyzed using LC-MS^E^ on a Waters G2-XS QToF spectrometer (Waters Corp., Milford, MA, USA) using a Supelco Ascentis Express C18 column (10 cm × 2.1 mm, 2.7 µm) in negative-ion mode ESI. The same extracts were also analyzed separately using flow injection analysis (FIA) in positive-ion mode ESI to measure neutral lipids that were not detected as negative ions. MNT annotations were based on searches of several databases (Metabolomics Workbench, Human Metabolome Database, METLIN). Confidence in metabolome annotation ranged from Metabolomics Standards Initiative categories of (1): (authentic standards matched retention times and mass spectra for common amino acids, and caffeine; (2): compounds matching database spectra for at least one characteristic fragment ion; (3): compounds annotated to a compound class; and 4: the majority of non-annotated features assigned as unknowns) [24]. Additional analysis details are provided in the Appendix A.

The software used for processing MNT data (Progenesis QI) provided an initial assignment of a Compound ID in the format of Retention Time mass, followed by a designation of n (indicating the mass is for the neutral molecule as judged by multiple adduct ions being detected) or m/z (indicating this is the mass of a detected ion). Retention times (in minutes) were presented as numerical values with two decimal places, and masses were numerical values reported with four decimal places. Since the analysis of MNTs involved multiple analytical approaches (negative ion/reversed-phase LC/MS, positive-ion/HILIC LC/MS, and positive-ion flow-injection analysis), an additional designation was added to the beginning of the Compound ID (nlp, plp, and slp, respectively), and to facilitate processing using SAS software (9.4), decimal places and slashes were changed to underscores. For example, the MNT initially reported in the positive-ion HILIC LC/MS data as 2.94_279.0142m/z was converted to plp2_94_279_0142m_z.

### 2.5. Statistical Preprocessing of MNT Data

Three types of MNTs have been measured: polar MNTs (plp), positive lipids (slp), and negative lipids (nlp). We applied three steps to improve the data: (1) control of batch effects, (2) ranking of MNTs into a maximum of five ranks, and (3) removal of MNTs with near-zero variances.

(1)Reducing batch effects: In some batches, serum samples from F0 and F2 generations were analyzed together for MNTs. In other batches, only F2 serum samples were analyzed. We observed that some F0 compounds may have been oxidized due to storage duration (approximately 30 years). Thus, some differences in MNTs between batches reflect the variations in MNTs between F0 and F2 resulting from storage time, rather than pure batch effects. Removing such batch effects using the ComBat method eliminates differences in MNTs between F0 and F2 generations [25], due to, for instance, oxidation. Thus, batch effects were estimated based on signals for stable isotope-labeled internal standards added to each serum specimen at constant amounts, with most being detected exclusively in the polar fractions. We identified five negative-ion lipids considered stable to auto-oxidation (annotated with retention times and masses by Progenesis QI software and manually in parentheses) as 17.29_804.5762m/z (PC 34:1), 17.53_785.6000n (PC 36:2), 16.75_781.5628n (PC 36:4), 16.28_702.5676n (SM (d34:1)), 18.49_812.6716n (SM (d42:1)) and six stable positive-ion lipids (0.28_814.6822m/z (TG 48:5), 0.28_822.7629m/z (TG 48:1), 0.28_846.7610m/z (TG 50:3), 0.28_369.3562m/z (cholesterol ester fragment ion), 0.28_820.7478m/z (TG 48:2), 0.28_872.7728m/z (TG 52:4). These stable lipids have low degrees of unsaturation, were judged not to be affected by oxidation owing to the lack of detected oxidized forms, and can be used to check pure batch effects. Using these stable MNTs, factor analyses were conducted that provided two important principal components, which were prepared for potential adjustments. However, the principal components of these stable lipids were not significantly different across batches. Hence, for positive- and negative-ion analyses of the non-polar fractions (lipids), there was no need to adjust for batches. For polar MNTs, we measured signals of four internal standards: cotinine-*d*_3_ [1.22_180.1208m/z], [^13^C_3_]caffeine [1.05_215.1008n], valine-*d*_8_ [6.94_125.1291n], and phenylalanine-*d*_5_ [5.71_170.1102n] and identified two principal components (factors), which were related to batches. Appendix A shows that factor 1 in batch 19, 24, 29, and 15 deviates from the remaining batches, and so does factor 2 for batches 1–9. To mark these, two dummy variables (combined to one variable) were used (Appendix A: dummy 1: batches 19, 24, 29, and 15; dummy 2: batches 1–9), which capture differences among batches (Appendix A). The batch-group variable is adjusted as covariates in the statistical analyses.(2)Ranking MNTs: owing to the use of low thresholds for data import and peak detection, >50% of the MNTs had a relatively large (>30%) percentage of zeros. These zeros may include technical zero (e.g., values below detection limit or accidental technical errors in peak detection or thresholding) or biological zero (e.g., zero or near zero abundance). To deal with a large number of values below the detection limit with minimal sacrificing of relatively rare exposure markers, a quantile regression imputation of left-censored data (QRILC) approach can be applied for the imputation of left-censored missing, not at random data [26]. However, this approach may introduce a problem. MNT levels are often strongly right-skewed (severe outliers). If MNT measurements are used as continuous data, log transformation will be needed before implementing any normality-dependent statistical analyses. Though, if a large number (>30%) of zeros were imputed with a random small value, the log-transformation would exaggerate the influence of these small randomly imputed values and may bias the parameter estimation in the downstream analyses. Instead of imputing the excessive zeros (>30%) for a large number of MNTs (>50% of all MNTs), we ranked all MNTs based on signal abundances allowing up to five ranks (0/1/2/3/4) using PROC RANK in SAS by keeping all zeros still as zeros in ranking. This conservative approach also minimizes effects due to outliers.(3)Given that many MNTs had extremely low variances since these variables mainly consisted of non-detects (zeros), including these near zero-variance predictors into statistical models such as regression results in misleading findings or causes errors due to lack of variability. To avoid near zero-variance predictors, from the ranked data we removed MNTs which had more than 80% zeros [27,28].

### 2.6. Outcomes in the F1- and F2-Generations

In F1 participants, the International Study of Asthma and Allergy in Childhood (ISAAC) questionnaire was used to obtain information regarding asthma at 10, 18, and 26 years [29]. Asthma was defined as “physician-diagnosed asthma” and “wheezing or whistling in the chest in the last 12 months” or “current treatment for asthma”. Skin Prick Tests (SPT) at ages 4, 10, and 18 years were evaluated using 11 common allergens (house dust mite, cat dander, dog dander, grass pollen mix, tree pollen mix, *Alternaria alternata*, *Cladosporium herbarium*, cow’s milk, hen’s egg, peanut, and cod). Being SPT-positive to one or more of the 11 allergens (weal diameter ≥3 mm) was treated as being positive for SPT. Total immunoglobulin E (IgE) at ages 10 and 18 years was assessed using Immunocap (Phadia, Uppsala, Sweden), designed to measure IgE between 2.0 to 1000 kU/L [30]. Forced exhaled Nitric Oxide (FeNO) was determined (Niox mino, Aerocrine AB, Solna, Sweden in parts per billion (ppb)) according to American Thoracic Society (ATS) guidelines [31] at ages 18 and 26 years. Lung functions were measured using a KoKo Spirometer and Software with a portable desktop device (both PDS instrumentation, Louisville, KY, USA) according to ATS guidelines [32]. Measurements of Forced Vital Capacity (FVC), Forced Expiratory Volume in one second (FEV1), and forced mid-expiratory flow (FEF_25–75%_) were performed at ages 10, 18, and 26 years. The ratio of FEV_1_/FVC was calculated.

Again, in the F2 generation, the International Study of Asthma and Allergy in Childhood (ISAAC) questionnaire was used to obtain information at 3, 6, 12, 24, 36 months, and between 6–7 years. Asthma was defined as “wheezing or whistling in the chest in the last 12 months”, “dry cough at night apart from a cough associated with a cold or chest infection”, or “child wheeze between colds or chest infections” and “current treatment for asthma (bronchodilators or inhaled corticosteroids)” or “physician diagnosed asthma”. SPTs were applied at ages 12, 36 months, and 6–7 years. Total serum IgE was assessed at age 6–7 years using Immunocap (Phadia, Uppsala, Sweden), designed to measure IgE between 2.0 to 1000 kU/L [30]. FeNO (Niox mino, Aerocrine AB, Solna, Sweden) was measured at 6–7 years according to American Thoracic Society (ATS) guidelines [31]. FeNO was determined (Niox mino, Aerocrine AB, Solna, Sweden) and lung function parameters at age 6–7 years were measured using a KoKo Spirometer and Software with a portable desktop device (both PDS instrumentation, Louisville, KY, USA) according to ATS guidelines [31,32]. The latter provided information on FVC, FEV_1_, their ratio, and FEF_25–75%_.

### 2.7. Covariates in the F1- and F2-Generations

In F1, information regarding sex and birth order was extracted from questionnaire data. Socio-economic status (SES) was defined based on household income, number of rooms, and maternal education [33]. Information on breastfeeding practices and total breastfeeding duration (relates to the number of weeks a mother breastfed her child regardless of the introduction of formula and/or solid food) and introduction of formula and solids were obtained through questionnaires answered by the mothers at the 1- and 2-year follow-ups. For our analysis, breastfeeding was used as a categorical variable (exclusive breastfeeding group, exclusive formula feeding group, and mixed feeding group) [34]. Active smoking status at ages 10, 18, and 26 years was recorded as “yes” if the participant was a current smoker. Second-hand smoke exposure at ages 10, 18, and 26 years was determined using questionnaire information obtained for tobacco smoke exposure from mother, father, others, or outside home. Maternal age at birth was calculated due to birth record data. For lung function parameters, we additionally adjusted for the height of the participant. To evaluate the differential contribution of age at outcome assessments (10, 18, and 26 years) on the effect of MNTs, “age” was included in the statistical models as an adjusting factor or as an interaction term with MNTs. Finally, for polar MNT differential, variation among batches was adjusted by a batch variable.

In F2, we also stratified for sex of the child and adjusted for birth order (parity), and the level of maternal smoking during pregnancy. For all lung function parameters, we additionally took the height of the child into consideration.

### 2.8. Analyses of Associations between MNTs and Multiple Allergic and Respiratory Outcomes

To compare whether the analytical sample represents the two birth cohorts, the F1-or the F2-generation, one-sample proportion tests were used for categorical levels. For normally distributed continuous variables, one-sample t-tests and for non-normal distribution, Wilcoxon signed rank tests were applied.

In the F1 generation, to identify MNTs related to allergic and respiratory outcomes (measured at different ages), we used two steps: first, a screening of informative MNTs, and second, statistical analyses with adjustment for potential confounding factors. In the first step, for each outcome measured at each time point (i.e., age), an R package, *ttScreening*, was applied to screen all MNTs in the F0 maternal serum separately for potential associations with outcomes in the F1 generation [35]. Polar MNTs (abbreviation PLP), and lipids measured in negative- (NLP) and positive-ion (SLP) modes, respectively, were analyzed separately. ttScreening is a screening approach utilizing training and testing samples to filter out uninformative MNTs. MNTs showing statistical significance in at least 50% of randomly selected training and testing datasets were selected as potentially outcome-associated MNTs. These informative MNTs were then considered for further analysis. In the second step, generalized logistic, linear, or log-linear regression models with repeated measurements were conducted to evaluate the association of the selected MNTs with each outcome adjusted for potential covariates (confounders) described above. The age of assessment was included as a categorical variable (time). Additionally, interaction effects of MNTs and time of assessment (i.e., age when outcomes were observed) were assessed for each outcome. For MNTs not showing interaction effects, their main effects are presented.

The following five steps were used to examine the biological and statistical reliability of the associations. First, we checked whether one MNT was related to two or more different outcomes in consistent directions. Second, we determined whether the same association was seen in boys and girls. Third, we reviewed whether correlated MNTs (Spearman correlation >0.7) showed similar associations with outcomes in the same directions. Fourth, we applied multiple testing adjustment separately for associations detected for polar MNTs, lipids measured in negative-ion, and lipids measured in positive-ion mode using the false discovery rate method. Fifth, MNTs that fulfilled an FDR-adjusted *p*-value of 0.05 were tested for replication in the F2-generation. In this study, to be on the conservative side, we used FDR adjusted *p*-value ≤ 0.05 in the discovery analyses (F1), but raw *p*-value ≤ 0.2 in the replication analyses (F2). For justification see the Appendix A.

## 3. Results

The occurrence of multiple allergic and respiratory outcomes in the total F1 and F2 cohorts and their respective analytical samples, with few exceptions, suggest that the analytical sample is representative of the complete cohort (Table 1a,b). Exceptions are lung function parameters in F2-boys at six years which are lower in the analytical samples.

The original MNT data for the F0-generation include 2286 negative-ion lipids, 59 positive-ion lipids (none of which were detected in the negative-ion data), and 6331 polar MNTs (positive-ion mode). Of these, retaining MNTs having fewer than 80% zeros reduced the size of the datasets to 1585 (69%) negative-ion lipids, all 59 positive-ion lipids, and 5264 (83%) polar MNTs. Associations showing increased risks are presented in Table 2a,b, separated for female and male participants. Accordingly, associations conveying protective effects are shown in Table 3a,b.

Since the analysis of MNTs involved multiple analytical approaches (negative ion/reversed phase LC/MS, positive-ion/HILIC LC/MS, and positive-ion flow-injection analysis), an additional designation was added to the beginning of the Compound ID (nlp, plp, and slp, respectively), and to facilitate processing using SAS software, decimal places and slashes were changed to underscores. For example, the MNT initially reported in the positive-ion/HILIC LC/MS data as 2.94_279.0142m/z was converted to plp2_94_279_0142m_z.

Since the analysis of MNTs involved multiple analytical approaches, an additional designation was added to the beginning of the compound ID (nlp, plp, and slp respectively), and to facilitate processing using SAS software, decimal places and slashes were changed to underscores. For example, the MNT initially reported in the positive-ion/HILIC LC/MS data as 2.94_279.0142m/z was converted to plp2_94_279_0142m_z.

*First*, we checked for MNTs showing associations with more than one outcome. For female offspring (Table 2a and Table 3a), a total of 47 associations between MNTs and allergic/respiratory outcomes were identified, representing 45 individual maternal MNTs. Of the latter, 33 MNTs are linked to a higher risk and 12 convey a protective effect. Three substances had associations with two outcomes each. Higher risks for both SPT and FeNO were linked to plp1_45_242_1558m_z (annotated as desmethyldiphenhydramine, a metabolite of a common antihistamine/sedative, SPT and FeNO increase, Table 2a, Appendix A). Another dual link was seen for plp6_96_202_0376m_z (annotated as 4-amino-2-methyl-5-phosphooxymethylpyrimidine, an intermediate in vitamin B_1_ (thiamine) biosynthesis and attributed to gut microbial action) and FVC and FEV_1_ (FVC decrease, FEV_1_ decrease, Table 2a, Appendix A). Higher levels of this compound may suggest its inefficient conversion into thiamine. Although serum levels of thiamine did not show significance in this study, we assumed >100-fold lower mean levels measured in F0 sera relative to F2 sera can be attributed to thiamine decomposing during 30+ years of serum storage and not presenting a useful compound for associations in F0 [36,37]. In female participants, a discordant link (SPT decrease at 18 years—IgE increase at all ages, Table 2a and Table 3a, Appendix A) was found for plp10_25_189_1597m_z, annotated as amino acid derivative N6,N6,N6-trimethyl-L-lysine (C_9_H_20_N_2_O_2_).

For male offspring (Table 2b and Table 3b), 35 associations showed increased risks with a nominal *p*-value of 0.05 representing 35 MNTs; 18 associations conveyed a protective effect representing 14 MNTs. Regarding the latter, four MNTs had consistent associations with more than one outcome, all among the protective MNTs (Table 3b). Three substances found in maternal serum during gestation were protectively related to higher FVC and FEV1 in offspring: (a) plp2_92_351_0460m_z, an unknown MNT assigned the chemical formula C_6_H_16_N_4_O_9_P_2_ which matches the formula of phospholombracine, a metabolite found in earthworms but without precedent in human sera (both FVC and FEV1 increased at 26 years, Table 3b, Appendix A); (b) plp2_94_279_0142m_z, 3,5-dimethoxy-4-(sulfooxy) benzoic acid (a sulfate conjugate of the polyphenol syringic acid, which is present in some foods but also a product of gut microbial metabolism) (both FVC and FEV1 increased at 26 years, Table 3b, Appendix A); and (c) plp2_94_799_4893m_z, a substance annotated as the phosphatidyl-ethanolamine lipid PE 38:8, a polyunsaturated phospholipid related to FVC and FEV1 (both FVC and FEV1 increased at 26 years, Table 3b, Appendix A). In addition, slp0_28_326_3818m_z, annotated as didecyl dimethylammonium (DDAC), an antiseptic substance, was associated (protectively) with a higher ratio of FEV1/FVC and to higher FEF_25–75%_ values (Table 3b, Appendix A). *Second*, regarding agreements between male and female offspring, no common MNTs were found.

In the *third* step, we investigated the correlations between MNTs. The heat plots in Appendix A show the Spearman correlation between individual MNTs. Regarding Spearman correlation >0.7, one and three groups of correlated MNTs were identified in female and male offspring, respectively (Appendix A). All correlated MNTs show similar associations with outcomes in the same directions. In females, the one group of cholesterol derivatives and diacylglycerol is related to a higher risk of a positive SPT. In male offspring, three groups of correlated MNTs and one pair of MNTs indicated higher risks (Appendix A). One group shows a protective effect of correlated MNTs for FVC and FEV_1_ (Table 3b, plp2_92_351_0460m_z related to FVC, and FEV_1_, Appendix A). Regarding both outcomes, this substance is correlated with plp2_94_279_0142m_z (Appendix A), a sulfate conjugate of the polyphenol syringic acid, which in unconjugated form, is present in some foods but is also a product of gut microbiome activity.

*Fourth*, we found two MNTs with low *p*-values (FDR adjusted *p*-value ≤ 0.0005) in association with allergy outcomes, both in female offspring (Table 2a). These are plp5_44_554_2593m_z, annotated as the steroid metabolite aldosterone 18-glucuronide, and plp8_04_132_0726m_z, an unknown substance. Both are related to a higher risk of a positive SPT (Table 2a, Appendix A).

*Fifth*, in the last step, we tested all MNTs with FDR-adjusted *p*-values of ≤0.05 in the F1 generation for replication in the F2 generation (*p*-value of ≤ 0.2). In addition, it was required that the associations between MNTs and respiratory outcomes show the same directions (either risk or protection). Among the 33 MNTs related to an increased risk of respiratory and allergic outcomes in female F1-participants, 2 were replicated in F2 (Table 4). Of the 35 MNTs significant in males of the F1-generation, 7 were replicated. Among the 12 protective MNTs in female participants of the F1-generation, 3 were also found significant in the F2-generation. Of the 14 protective MNTs in male F1 participants, 9 were replicated (Table 5). Of the nine MNTs, two were associated with two different lung function parameters (FVC and FEV_1_), one with an unknown (non-annotated) compound (plp2_92_351_0460m_z).

Among MNTs that were risk factors in girls, plp1_60_369_2086m_z, benzyl (2-ethylhexyl) phthalate, a phthalate plasticizer, increased FeNO in the F1 generation (Appendix A) and was replicated in F2 (Table 4). Similar results were found for plp1_45_242_1558m_z, an over-the-counter (OTC) antihistamine/sedative metabolite (Appendix A, Table 4). In male F1 and F2 offspring, plp0_95_237_1019m_z, glycosminine, a quinazoline alkaloid found in the plant family Rutaceae which includes ornamentals and citrus foods, was associated with a higher risk of asthma (Appendix A). IgE in male F1 and F2 offspring was increased with higher exposure to a bile acid glucoside (plp5_70_588_3739m_z) and a dipeptide of branched chain amino acids annotated as one of Ile-Val; Val-Ile; Leu-Val; Val-Leu (plp5_90_231_1698m_z) (Appendix A, Table 4).

In males, four MNTs were associated with FeNO in F1 and F2 participants. These are one unknown substance, tryptamine (a tryptophan metabolite), prenyl glucoside (a hemiterpenoid glycoside associated with citrus), and polyunsaturated triacylglycerol TG 60:8, a potential precursor of bioactive eicosanoids (Table 4, plp1_57_497_2341m_z in Appendix A, plp1_05_161_1074m_z, plp8_60_266_1595m_z, slp0_28_976_8400m_z in Appendix A).

Regarding MNTs that convey protective effects in female participants (Table 5), there is a lower risk of SPT positivity related to two MNTs in F1 and F2 (plp10_25_189_1597m_z and plp6_13_119_0928m_z (Appendix A), and a higher FEF_25–75%_ effect related to one MNT with unknown annotation (nlp16_65_861_5483m_z, Appendix A). Plp10_25_189_1597m_z, annotated as amino acid derivative N6,N6,N6-trimethyl-L-lysine (C_9_H_20_N_2_O_2_), has endogenous and dietary sources, abundant in some meats, seafood, and eggs, and more recently reported to be prevalent in many vegetables [38]. This metabolite serves as an important precursor of carnitine, a key metabolite involved in the transport of fatty acids to mitochondria for biochemical energy generation.

In male participants, regarding the lung function parameters FVC, FEV_1_, and FEF_25–75%_, nine MNTs found in F0-maternal serum and cord serum of F2-newborns were associated with increased lung function (Table 5). Since two of the nine MNTs were each related to two different lung function parameters, there were eleven associations (Appendix A). The nine MNTs include a fully saturated triacylglycerol (TG 60:0) and the sulfate conjugate of syringic acid, a polyphenol. The latter is correlated (Spearman correlation ≥0.7) with two unknown substances, plp2_94_295_0654m_z and plp2_92_351_0460m_z, and a medium chain ketoacid (Appendix A).

## 4. Discussion

Our approach aimed to capture a variety of serum MNTs using a systematic and untargeted analysis of a heterogeneous group of compounds applying low peak detection thresholds. The results of this untargeted approach to parallelly assess multiple MNTs for associations with respiratory and allergic markers and their replication provide comparative information on their importance. In the F0 generation, we detected 33 MNTs (35 associations) in maternal serum collected at the end of the pregnancy that constituted a higher risk for respiratory and allergic outcomes in female offspring (Table 2a) and 35 MNTs related to a higher risk in males (Table 2b). In each table, 11 were unknown compounds. Accordingly, we identified 12 MNT in F0 in female and 15 in male offspring (18 associations) that suggested a protective association (Table 3a,b). In female participants, seven MNTs could not be annotated (unknown substances), and six in male participants. Among MNTs having a higher risk for allergic/pulmonary outcomes, nine were replicated in F2 with one annotated as unknown (Table 4). For MNTs constituting potential protection, 12 MNTs (14 associations) were found with 4 compounds annotated as unknowns (Table 5).

Since we detected medications used by the mother during pregnancy (indicative of maternal disorders) that were related to offspring respiratory and allergic disorders (indication bias), we believe that our systematic and untargeted analysis of a heterogeneous group of compounds was able to identify a large number of important chemical compounds during pregnancy with low peak detection.

The untargeted approach constitutes a challenge, since multiple substances were unknowns, not found in databases, and could only be annotated based on molecular masses, isotopolog abundances, fragment ion masses, and expected chromatographic retention times. Despite the complexities of the untargeted technique, this approach has meaningful advantages. First, the untargeted approach provides information on the relative importance of individual MNTs, which cannot be achieved with targeted candidate MNT approaches, since the analyses of candidates focus on a few targets but neglects the wider picture. Interestingly, MNTs that we detected previously in candidate-like approaches with DNA-methylation did not show significant associations with any respiratory or allergic outcomes assessed in this study including cotinine (a marker of exposure to tobacco smoke) and acetaminophen metabolites [39,40]. Second, the untargeted approach offers novel insights into MNTs that were not yet considered candidate substances in association with pulmonary or allergic outcomes. These compounds were discovered in F0-F1 and replicated in F2: a phthalate plasticizer, an OTC antihistamine/sedative metabolite, a quinazoline alkaloid attributed to citrus consumption, a bile acid metabolite, tryptophan metabolites, a hemiterpenoid glycoside, triacylglycerols, hypoxanthine, oxidized phosphatidyl-ethanolamine, and sulfate conjugate of the polyphenol syringic acid (Table 4 and Table 5).

In female offspring, in the discovery and the replication cohort, we observed increased FeNO levels in association with phthalate exposure namely benzyl (2-ethylhexyl) phthalate. Phthalates are widely used in many substances including plastics, personal care products, and vinyl floors [41]. FeNO is an indicator of airway inflammation related to asthma. Previous studies have shown an association between increased exposure to phthalates with a higher risk of asthma and poor lung function parameters in children [41,42,43,44]. Phthalates have been also reported to increase oxidative damage to airway epithelial cells, leading to easy entry and uptake of allergens by dendritic cells [45]. The resulting disruption of the epithelial barrier and introduction of allergens to the immune system disturbs the balance of Th1/Th2 towards production of Th2 cells and its associated pro-allergic cytokines [38]. However, not all phthalates act on the host by common mechanisms, and the biological effects of the specific phthalate identified in this study, benzyl (2-ethylhexyl) phthalate, have not been extensively studied.

We also detected an association between desmethyldiphenhydramine, a metabolite of diphenhydramine, and a higher risk of SPT positivity in female offspring, both in the F1- and F2-generations. Diphenhydramine (or Benadryl) is an antihistamine drug with a broad usage in allergic and dermatological conditions such as atopic dermatitis [46]. It is used in pregnancy as an anti-pruritic or anti-emetic agent [46]. The association observed in our investigation could be due to an indication bias, meaning that mothers prone to allergic symptoms were more likely to use antihistamines during pregnancy. Then, their offspring might be also more likely to inherit allergy-associated genes and show a positive SPT later in life, which in turn can result in a spurious association between diphenhydramine and SPT positivity.

The quinazoline alkaloid annotated as glycosminine measured at birth was found to increase the risk of asthma in male offspring, both in the discovery and replication samples. Quinazolines are a broad family of compounds that are ubiquitously used in different medications including anti-virals, anti-fungals, anti-malarial agents, anti-hypertensives, and anti-inflammatory drugs [47]. This particular compound is a natural product reported in the Indian medicinal plant *Glycosmis arborea* [48], a member of the plant family Rutaceae, which includes all of the common citrus fruits. However, some quinazoline derivates also provide bronchodilator activity [49]. Hence, the reported association could also be due to an indication bias. In this case, mothers with asthmatic symptoms used bronchodilators during pregnancy and their offspring were more likely to develop asthma. This then results in a spurious association between quinazoline and asthma in offspring.

We observed that prenyl glucoside was associated with increased FeNO levels in male offspring, discovered in the F1- and replicated in the F2-generation. The hemiterpenoid glycoside has been isolated from flower buds of satsuma mandarin (mandarin orange) [50]. Since the fruit develops from the flower, it may be expected that such compounds are also present in this, and perhaps other citrus fruit consumed during pregnancy.

Our data show that bile acid metabolite (3α-[(β-D-glucopyranosyl)oxy]-7α,12α-dihydroxy-5β-cholanic acid, a glucoside metabolite of the primary bile acid cholic acid) measured at birth is a risk for higher IgE levels in male offspring at 10 and 18 years. Bile acid salts are released into the duodenum as end-products of hepatic cholesterol metabolism [51]; a liver glucosyltransferase catalyzes the formation of bile acid glucosides [52]. Bile salts exert antimicrobial effects in the intestinal lumen and are linked to gut microbiota [51]. They are further metabolized by the gut microbiome to species distinct from the host [51,53,54]. This interaction between human host and gut microbiota has been implicated in the development of certain diseases such as asthma [53]. Bile acid metabolites produced by gut microbiota act as messenger molecules. Certain metabolites of bile acids have been shown to affect regulatory T cell differentiation hindering tolerance development to oral antigens [55]. The association we observed between the bile acid metabolite and IgE may be explained by disturbed gut microbiota and resulting food allergy and IgE production [56].

We found that tryptamine, an indole metabolite of tryptophan, may act as a risk factor for higher FeNO levels in male offspring, discovered in F1- and replicated in F2- offspring. Tryptophan is an essential amino acid with a complex metabolism. A major pathway of tryptophan metabolism leads to the production of kynurenine derivatives by indoleamine 2,3-dioxygenase-1 (IDO-1) in antigen-presenting cells and other cells of the immune system [57]. IDO breaks down tryptophan as part of an antiproliferative strategy of T cells to avoid their overactivation [58]. Prior findings support the role of tryptophan metabolites in the pathogenesis of asthma [58]. The fecal microbiome is another major contributor to tryptophan metabolism [57], particularly contributing to tryptamine formation [59]. Tryptamine is one of several bacterial metabolites of tryptophan recognized as aryl hydrocarbon receptor (AhR) ligands, presenting a potential mechanism for regulating intestinal immunity [60]. Our findings seem to conflict with the prior investigation of existing asthma and FeNO showing lower tryptophan metabolites in asthmatic children [61,62]. However, these studies investigated tryptophan and its metabolites in children with existing asthma (potential of inverse causation) whereas our study linked tryptophan metabolite measured in maternal serum at birth or cord serum with the development of increased FeNO, as a marker of asthma, in the offspring later in life.

Two serum triacylglycerols were discovered in F0-maternal serum and linked to male F1-offspring FeNO (18 and 26 years, increased risk) and FVC (10, 18, 26 years, protective association) and replicated in male F2-offspring (cord serum and FeNO and FVC at 6–7 years). A polyunsaturated form slp0_28_976_8400m_z (TG 60:8) was associated with increased FeNO, reflective of inflammation, and may serve as a precursor of pro-inflammatory eicosanoid lipid mediators. The number of acyl carbons in fatty acids (60) would be consistent with eicosanoid fatty acid groups, and the large number of double bonds (eight) provides evidence of polyunsaturation consistent with eicosanoids, which would explain the higher FeNO levels due to inflammation. Another, however, fully saturated triacylglycerol slp0_28_992_9463m_z (TG 60:0), was associated with a protective effect resulting in higher FVC in F1- and F2-males. Regarding FeNO and FVC, the opposite associations of polyunsaturated and fully saturated triglycerides need more evaluation.

We observed a protective effect for a sulfate conjugate of a polyphenol in association with FVC in male offspring, both in the discovery (F1) and replication (F2) groups. Polyphenols are naturally occurring antioxidants abundant in fruits and vegetables. Two cross-sectional studies showed a positive association between dietary polyphenol intake and lung function parameters [63,64]. Polyphenols are believed to exert their protective effects through their roles as anti-oxidants and by reducing inflammation [63].

Another protective association of a polyunsaturated phosphatidylglycerol (PG 40:9) was seen for FEV1 and FVC, suggesting a lung-protective effect. Phosphatidylglycerol (PG) is a class of phospholipids and this particular PG, with 40 acyl carbons and 9 double bonds, is consistent with containing esters of 2 ω-3 fatty acids: linolenic acid (ALA, 18:3n3) and docosahexaenoic acid (DHA, 22:6n3). PGs are acidic phospholipids that typically comprise 7–15% of phospholipids in pulmonary surfactant [65]. The lipid class PG has been reported as a minor component of human blood [66], has a low abundance in membranes of mammalian tissues relative to bacterial and plant membranes, and has been documented to inhibit TNF-α production following lipopolysaccharide challenge to macrophages [67]. Although the relationships between levels of PG in serum and lung surfactant are unclear, decreases in surfactant lipids including PG have been associated with reduced lung function in COPD patients [68]. Our observation may also be explained by age-dependent changes in the phospholipid composition of surfactants [69].

The chemical 7-amino-4-hydroxy-2-naphthalenesulfonic acid (HMDB0243485, Table 5) is a likely intermediate in the production of azo dyes or a breakdown product of dyes. It has been included in the Blood Exposome Database (bloodexposome.org) and is reported to be a skin irritant under UN GHS Classification. However, after breakdown by the enzyme laccase, the phenolic and arylamine functionalities are expected to confer antioxidant properties [70]. The enzyme laccase is common in bacteria and may be present in the gut microbiome. Noteworthy, this compound is not an azo dye, and we did not detect any intact azo dyes related to respiratory and allergic outcomes. More research is needed to understand the role of this naphthalene sulfonic acid for these outcomes.

Regarding plp2_94_137_0413n and plp2_94_177_0647n, both were annotated as signals of hypoxanthine and were protectively related to FEV1 in male participants of the F1 and F2 generations (Table 3b and Table 5). The chemical assessment showed there is a third MNT signal related to these two, namely, plp2_94_136_0413n, also corresponding to hypoxanthine. All three MNT signals are attributed to a single MNT compound, hypoxanthine, but represent different isotopologs or solvent adducts that were unexpectedly not combined by the Progenesis QI software. However, plp2_94_136_0413n did not pass the screening process and thus was not further checked in linear mixed models adjusting for confounders. Nevertheless, all three MNT signals are highly correlated and share a common chromatographic retention time, suggesting they come from a single compound (Spearman correlation in male F1 participants: plp2_94_137_0413n and plp2_94_177_0647n 0.87648 (*p* < 0.0001); plp2_94_137_0413n and plp2_94_136_0384n 0.99648 (*p* < 0.0001)). Hence, when in F1 male participants the three potential hypoxanthine signals were combined, their association with FEV1 repeatedly measured at ages 10, 18, and 26 years and controlled for confounders was statistically significant with a *p*-value lower than for the original measured MNTs (*p* = 0.0095). This suggests that hypoxanthine is associated with a protective effect on FEV1 in male participants. Hypoxanthine is involved in purine biosynthesis and nucleotide metabolism and is a precursor of uric acid via metabolic steps that form reactive oxygen species. Liang et al. [71] in a cross-sectional analysis compared diseased (asthma and COPD) and healthy adults. Contrary to our findings, the authors suggested that hypoxanthine constitutes a risk. However, other than this cross-sectional comparison, our study has a clear time order from birth to later in childhood or adulthood and assesses longitudinal associations. In addition, a recent cohort study with cystic fibrosis patients comparing pulmonary exacerbations (hospitalization with at least a 10% decrease of FEV1) compared to normal outpatient clinic visits showed that the participants had significantly lower hypoxanthine levels when they experienced exacerbations [72]. The authors discussed that a decreased hypoxanthine concentration may be secondary to its increased conversion to uric acid during an exacerbation, generating superoxide and hydroxyl radicals and resulting in cellular damage. Nevertheless, our findings suggest that an increased hypoxanthine level (measured in maternal blood at the end of the pregnancy and in cord blood) may be primary, thus protective, against the reduction in FEV1 later in lives of the male offspring. This view is also supported by experimental studies [73,74,75,76]. Given this combined effect in male participants, we additionally checked a potential association with FEV1 in female F1 participants but found none. In addition, since uric acid (plp6_60_169_0372m_z) is a metabolite of hypoxanthine, uric acid was analyzed in male and female F1 participants, but no associations were found with lung function markers (FVC, FEV1, their ratio, and FEF_25–75%_).

We did not find any reference for three known polar MNTs (an amino acid derivative, an aryl thioether, and a medium chain ketoacid, Table 4 and Table 5) and four unknown substances that were discovered in F0 maternal blood to be associated with respiratory outcomes in F1 and replicated in the F2 generation.

A striking interpretation of our findings is that many significant associations are compounds (e.g., CE 22:6, the cholesterol ester of the omega-3 fatty acid docosahexaenoic (DHA)) that reflect dietary inputs (fish consumption) as well as the numerous MNTs that are arisen from the activity of the gut microbiome such as bile acid metabolites and tryptamine.

In Table 2a,b and Table 3a,b we presented all substances discovered in F0 mothers related to respiratory and allergic outcomes in F1 offspring. Only a part of these was replicated in F2. However, this information may stimulate other investigators to test whether some of these substances are replicable in other cohorts.

Our study has strengths and limitations worth mentioning. Regarding the strengths of the study, we assessed metabolic links of several pulmonary/allergy outcomes including asthma, SPT, FeNO, IgE, and lung function parameters in an untargeted approach separated for female and male study participants. The untargeted metabolome-wide approach provides an opportunity to look at a wide range of MNTs. We examined multiple biological links associated with MNTs and multiple outcomes of a single MNT. We checked whether associations discovered in the F1-offspring based on F0-maternal sera could be replicated in F2-offspring based on F2 cord sera. This idea of discovery (in F0–F1) and replication (F2) in two consecutive generations provides both limitations and opportunities. We have 583 F1- and 230 F2-participants. A total of 75 families provided both F1- and F2-participants. Hence, the discovery and replication samples are only partially independent. Statistically, a discovery and replication approach requires independent samples which is only partially fulfilled in our study. On the other hand, analyzing two generational samples, which are not totally independent, since F0-mothers were not excluded from the discovery if their F1-offspring contributed to the F2-generation, may be biased in favor of MNTs that remain stable in families over generations. On the other side, a limitation of the discovery and replication approach in two subsequent generations is that MNTs can change over time. Some MNTs may occur in higher concentrations in the first generation but cannot be detected in the next and vice versa. Thus, in independent cohorts these MNTs are conceptionally excluded from replication. Another limitation of the current study is that MNTs were measured only once in maternal and cord serum. We suggest that future studies take serum samples at different time points to assess the variation of these MNTs over time. Overall, the respiratory and allergic outcomes of the analytical samples in F1 and F2 male and female participants were not different from those in the total cohort. Exceptions are lung function parameters in F2-boys at six years. However, this is not a limitation for the assessment of associations between MNTs and respiratory outcomes. The associations are still valid since the MNTs are distributed independently over these markers. Nevertheless, the generalization of these links in F2 boys for the total sample is limited.

A substantial number of detected serum MNTs did not match any metabolite or mass spectrum database entries and remained annotated as unknowns. Our efforts focused on annotating those MNTs that showed significant association with outcomes. In some cases, MNTs were left annotated as unknowns because many isomeric metabolite database entries matched but the analytical data did not allow further refinement to a single compound or limited number (usually <6) of options.

## 5. Conclusions

Based on a metabolome-wide untargeted approach, our study provides unique findings on associations of serum levels of metabolites, nutrients, and toxins with respiratory and allergy-related outcomes. We detected novel components harboring protective or harmful effects related to various allergic/pulmonary outcomes. Future studies should focus on phthalate plasticizers, bile acid and tryptophan metabolites, triacylglycerols, hypoxanthine, and a sulfate conjugate of the polyphenol syringic acid. In addition, two compounds related to citrus fruits, namely, quinazoline alkaloid and hemiterpenoid glycoside, and multiple metabolites including triacylglycerols that originate from our diet and/or the gut microbiome need our attention. Our findings should trigger preventive studies, addressing both avoidance of potentially harmful compounds and trials of potentially protective MNTs.

## Figures and Tables

**Table 1 metabolites-13-00737-t001:** (**a**). Comparison of outcome variables between the IOW cohort and the analyzed subsample, stratified by sex in the F1-generation. (**b**). Comparison of outcome variables between the IOW cohort and the analyzed subsample, stratified by sex in the F2-generation.

**(a)**							
**Outcome Variables**	**Age (yrs)**	**Female Participants**			**Male Participants**		
		**Complete Cohort, n = 721** ***n*** **(%)**	**Subsample with MNT,** ** n = 298, *n* (%)**	** *p* ** **-Value**	**Complete Cohort, n = 735,** ***n*** **(%)**	**Subsample** **with MNT** **n = 287, *n* (%)**	***p*-Value**
Asthma	10	672 83 (12.4%)	290 36 (12.4%)	0.98	696 118 (17%)	284 58 (20.4%)	0.20
	18	659 128 (19.4%)	285 57 (20%)	0.84	646 103 (15.9%)	266 52 (19.6%)	0.19
	26	560 97 (17.3%)	253 48 (19%)	0.57	470 63 (13.4%)	203 32 (15.8%)	0.42
Skin prick test (SPT) positivity	4	488 82 (16.8%)	218 42 (19.3%)	0.45	486 109 (22.4%)	215 56 (26.1%)	0.30
	10	518 119 (22.9%)	256 61 (23.8%)	0.79	509 159 (31.2%)	256 85 (33.2%)	0.58
	18	444 159 (35.8%)	219 78 (35.6%)	0.96	397 189 (47.6%)	196 101 (51.5%)	0.38
IgE (kU/L, geometric mean)	10	474 1.87 (0.74)	238 1.88 (0.73)	0.88	479 1.94 (0.75)	238 2 (0.75)	0.23
	18	235 1.9 (0.7)	141 1.88 (0.70)	0.68	221 2.03 (0.76)	135 2.04 (0.79)	0.87
FeNO (ppb, geometric mean)	18	435 1.19 (0.3)	212 1.19 (0.31)	0.87	387 1.34 (0.35)	188 1.37 (0.37)	0.22
	26	304 1.16 (0.3)	151 1.17 (0.29)	0.62	232 1.27 (0.32)	109 1.27 (0.32)	0.90
FVC (L)	10	493 2.24 (0.33)	244 2.22 (0.32)	0.24	488 2.35 (0.34)	245 2.33 (0.35)	0.48
	18	443 3.96 (0.53)	219 3.97 (0.55)	0.89	395 5.35 (0.72)	196 5.33 (0.73)	0.72
	26	311 4.24 (0.54)	156 4.27 (0.54)	0.43	236 5.85 (0.82)	108 5.78 (0.85)	0.39
FEV_1_ (L)	10	492 2.0 (0.29)	242 1.98 (0.29)	0.27	488 2.06 (0.3)	245 2.05 (0.32)	0.51
	18	443 3.47 (0.45)	219 3.49 (0.49)	0.66	396 4.62 (0.62)	197 4.6 (0.64)	0.72
	26	311 3.42 (0.43)	156 3.47 (0.43)	0.17	236 4.61 (0.72)	108 4.52 (0.72)	0.18
FEV_1_/FVC	10	492 0.90 (0.06)	244 0.90 (0.05)	0.23	488 0.88 (0.06)	245 0.88 (0.06)	0.63
	18	443 0.88 (0.07)	219 0.88 (0.07)	0.80	396 0.87 (0.07)	197 0.87 (0.07)	0.39
	26	311 0.81 (0.06)	156 0.81 (0.06)	0.46	236 0.79 (0.07)	108 0.78 (0.07)	0.39
FEF_25–75%_ (L)	10	493 2.48 (0.56)	244 2.47 (0.55)	0.71	488 2.38 (0.56)	245 2.37 (0.56)	0.80
	18	443 3.95 (0.87)	219 4 (0.93)	0.47	396 4.99 (1.16)	197 5 (1.2)	0.94
	26	311 3.44 (0.84)	156 3.52 (0.85)	0.28	236 4.37 (1.24)	108 4.2 (1.2)	0.15
**(b)**							
**Outcome Variables**	**Age (yrs)**	**Female Participants**			**Male Participants**		
		**Complete Cohort, ** **n = 339, *n* (%)**	**Subsample with MNT** **n = 118, *n* (%)**	***p*-Value**	**Complete Cohort,** **n = 268, *n* (%)**	**Subsample** **with MNT** **n = 112, *n* (%)**	***p*-Value**
Asthma	6	268 16 (5.97%)	112 11 (9.82%)	0.09	339 28 (8.26%)	118 14 (11.86%)	0.16
SPT	1–6	190 76 (40.0%)	88 43 (48.86)	0.09	260 100 (38.46%)	102 42 (41.18%)	0.57
IgE (kU/L, geometric mean)	6–7	69 0.228 (3.71)	65 0.206 (3.88)	0.51	72 0.11 (3.98)	69 0.025 (0.73)	0.93
FeNO (ppb, geometric mean)	6–7	69 7.2 (2.15)	38 6.78 (2.31)	0.66	85 8.06 (2.4)	40 9.19 (2.21)	0.31
FVC (L)	6–7	69 1.48 (0.36)	38 1.44 (0.25)	0.33	94 1.67 (0.56)	42 1.51 (0.29)	0.0008
FEV_1_(L)	6–7	69 1.34 (0.31)	38 1.3 (0.20)	0.22	93 1.47 (0.49)	42 1.32 (0.27)	0.0008
FEV_1_/FVC	6–7	69 0.91 (0.06)	38 0.91 (0.057)	0.78	93 0.88 (0.08)	42 0.88 (0.08)	0.97
FEF_25–75%_ (L)	6–7	69 1.74 (0.46)	38 1.71 (0.41)	0.64	94 1.7 (0.68)	42 1.51 (0.43)	0.01

Proportions and mean (SD) are shown for categorical and continuous variables, respectively. *p*-values are based on comparing the analyzed subsample with the complete cohort using one-sample chi-square or one-sample t tests for categorical and continuous variables, respectively.

**Table 2 metabolites-13-00737-t002:** (**a**). Metabolites, nutrients, toxins (MNTs) conveying higher risks for allergic and respiratory outcomes in female F1-offspring. (**b**). Metabolites, nutrients, toxins (MNTs) that convey a higher risk for allergic and respiratory outcomes in male F1-offspring.

**(a)**						
***Health Outcome* and Associated MNTs**	**Interaction with Time**	**Annotation**	**Compound Class**	**Chemical Formula**	***p*-Value ^#^**	**FDR Adjusted *p*-Value**
** *Asthma at 4, 10, 18, and/or 26 years* **						
plp1_52_182_1835m_z		Dicyclohexylamine	Organic amine	C_12_H_23_N	0.035	0.04
plp2_12_180_0878m_z		Dimethylguanine	Hypoxanthine	C_7_H_9_N_5_O	0.004	0.012
** *Skin prick test positivity at 4, 10, and/or 18 years* **						
plp0_79_858_7164m_z	yes	20:1-Glc-cholesterol	Cholesterol derivative	C_53_H_92_O_7_	0.0008	0.004
plp0_79_884_7325m_z	yes	22:2-Glc-cholesterol	Cholesterol derivative	C_55_H_94_O_7_	0.005	0.008
plp0_79_900_7263m_z	yes	22:2-Glc-cholesterol (ox)	Cholesterol derivative (oxidized)	C_55_H_94_O_8_	0.0004	0.003
plp0_80_710_5683m_z	yes	DG 42:8	Diacylglycerol, polyunsaturated	C_45_H_72_O_5_	0.02	0.02
plp0_80_834_7158m_z	yes	Unknown	Unknown		0.02	0.02
plp0_80_958_7302m_z	yes	Unknown	Unknown		0.01	0.02
plp0_82_581_4911m_z	yes	Unknown	Unknown		0.01	0.02
plp0_84_620_5956m_z		Cer (d18:2/22:0)	Ceramide	C_40_H_77_NO_3_	0.005	0.01
plp0_90_768_6313m_z		Unknown	Unknown		0.01	0.02
plp1_23_840_5319m_z		Unknown	Unknown		0.03	0.03
plp1_45_242_1558m_z		Desmethyldiphenhydramine	OTC antihistamine/ sedative metabolite	C_16_H_19_NO	0.003	0.01
plp5_44_554_2593m_z		Aldosterone 18-glucuronide	Aldosterone (steroid) metabolite	C_27_H_36_O_11_	0.00004	0.0004
plp7_09_116_0671m_z		Unknown	Unknown		0.002	0.01
plp7_09_241_0847m_z		Unknown	Unknown		0.005	0.01
plp8_04_132_0726m_z		Unknown	Unknown		0.00003	0.0005
plp8_90_242_0790m_z		N-Benzoylanthranilic acid	Food additive	C_14_H_11_NO_3_	0.0007	0.004
** *Immunoglobulin E levels* **						
plp4_49_138_0549m_z		Anthranilic acid	Tryptophan metabolite	C_7_H_7_NO_2_	0.001	0.004
plp10_25_189_1597m_z		N6,N6,N6-Trimethyl-L-lysine	Amino acid derivative, carnitine precursor	C_9_H_20_N_2_O_2_	0.002	0.004
plp2_03_100_0287m_z		Unknown	Unknown		0.004	0.005
plp5_66_110_0964m_z		1,2,5-Trimethyl-1H-pyrrole	Pyrrole	C_7_H_11_N	0.009	0.01
slp0_28_918_7488m_z		TG 56:9	Triacylglycerol (polyunsaturated)	C_59_H_96_O_6_	0.02	0.04
slp0_28_928_8330m_z		TG 56:4	Triacylglycerol	C_59_H_106_O_6_	0.04	0.04
** *Fractional exhaled nitric oxide (FeNO) at 10 and/or 18 years* **						
plp1_60_369_2086m_z		Benzyl (2-ethylhexyl)phthalate	Phthalate plasticizer	C_23_H_28_O_4_	0.002	0.01
plp3_46_202_0860m_z		1-Methyl-3-(2-oxo-propylidene)indol-2-one	Tryptophan metabolite (indole)	C_12_H_11_NO_2_	0.002	0.01
plp1_45_242_1558m_z		Desmethyldiphenhydramine	OTC Antihistamine/sedative metabolite	C_16_H_16_O	0.006	0.02
** *Forced vital capacity (FVC) at 10, 18, and/or 26 years* **						
plp1_05_188_0947n		Indolepropionamide	Tryptophan-metabolite	C_11_H_12_N_2_O	0.01	0.02
plp1_47_399_1925m_z		DiHDoHE	Oxidized Fatty acid (DHA)	C_22_H_32_O_4_	0.002	0.005
plp7_24_598_5132m_z		Unknown	Unknown		0.0005	0.002
plp6_96_202_0376m_z		4-Amino-2-methyl-5-phosphooxymethylpyrimidine	Aminopyrimidine metabolite	C_6_H_10_N_3_O_4_P	0.0002	0.002
plp6_18_273_1202m_z		Unknown	Unknown		0.0003	0.002
** *Forced expiratory volume in 1 s (FEV_1_) at 10, 18, and/or 26 years* **						
plp6_96_202_0376m_z		4-Amino-2-methyl-5-phosphooxymethylpyrimidine	Aminopyrimidine metabolite	C_6_H_10_N_3_O_4_P	0.0008	0.008
*FEV1/FVC ratio (none)*						
** *Forced mid-expiratory flow FEF_25–75%_ at 10, 18, and/or 26 years* **						
plp5_20_358_2207n		Asn-Ile-Ile or Gln-Val-Ile	Tripeptide	C_16_H_30_N_4_O_5_	0.003	0.01
plp2_03_235_1185m_z		Cyclo(His-Pro)	Dipeptide	C_11_H_14_N_4_O_2_	0.004	0.01
**(b)**						
***Health Outcome* and Associated MNTs**	**Interaction with Time**	**Annotation**	**Compound Class**	**Chemical Formula**	***p*-Value ^#^**	**FDR Adjusted *p*-Value**
** *Asthma at 4, 10, 18, and/or 26 years* **						
plp0_95_237_1019m_z		Glycosminine	Quinazoline alkaloid	C_15_H_12_N_2_O	0.004	0.012
** *Skin prick test positivity at 4, 10, and/or 18 years (none)* **						
** *Immunoglobulin E levels at 10 and/or 18 years* **						
plp5_65_195_0764m_z		4-Aminohippuric acid	Acyl glycine	C_9_H_10_N_2_O_3_	0.0002	0.003
plp4_71_187_1208n		Piperidione	Cough medicine Sedulon	C_9_H_15_NO_2_	0.0006	0.004
plp6_90_164_0686n		Fucose	Hexose	C_6_H_12_O_5_	0.0008	0.004
plp5_10_266_1161n		Unknown, numerous isomers	Unknown	C_14_H_18_O_5_	0.002	0.005
plp5_10_91_0535m_z		N-(Hydroxymethyl)urea	Urea derivative	C_2_H_6_N_2_O_2_	0.002	0.007
plp5_70_588_3739m_z		3α-[(β-D-Glucopyranosyl)oxy]-7α,12α-dihydroxy-5β-cholanic acid	Steroid metabolite; bile acid (cholic acid) glucoside	C_30_H_50_O_10_	0.002	0.005
plp5_90_231_1698m_z		Ile-Val; Val-Ile; Leu-Val;Val-Leu	Dipeptide	C_11_H_22_N_2_O_3_	0.002	0.005
plp5_10_91_0515m_z		N-(Hydroxymethyl)urea	Urea derivative	C_2_H_6_N_2_O_2_	0.004	0.005
plp0_84_1048_8866m_z		PG 54:0 (27:0/27:0)	Long chain saturated phosphatidylglycerol	C_60_H_119_O_10_P	0.004	0.007
plp5_70_502_2200m_z		Thamnosin	Coumarin	C_30_H_28_O_6_	0.009	0.01
plp2_19_578_4162m_z		PC(22:1/0:0)	Lyso phosphatidylcholine, monounsaturated	C_30_H_60_NO_7_P	0.01	0.012
plp0_81_244_2138m_z		Unknown	Unknown		0.01	0.012
plp5_43_116_0818m_z		Unknown	Unknown		0.01	0.012
plp0_80_563_4818m_z		Unknown	Unknown		0.04	0.045
** *Fractional exhaled nitric oxide (FeNO) at 18 and/or 26 years* **						
plp1_57_497_2341m_z		Unknown	Unknown		0.002	0.006
plp1_05_161_1074m_z	yes	Tryptamine	Tryptophan metabolite (indole)	C_10_H_12_N_2_	0.0004	0.004
plp1_67_168_1130m_z	yes	Unknown	Unknown	C_8_H_13_N_3_O	0.003	0.006
plp6_70_385_1611m_z	yes	Unknown		C_17_H_24_N_2_O_8_	0.01	0.02
plp3_16_370_2424m_z	yes	2,5,8,11,14,17-Hexaoxadocosan-22-oic acid	Polyether	C_16_H_32_O_8_	0.02	0.03
plp6_58_385_1611m_z	yes	Unknown		C_17_H_24_N_2_O_8_	0.02	0.02
plp7_39_311_1459m_z	yes	N-(Dimethylamino)methylene-9-((2-hydroxy-1-(hydroxymethyl)ethoxy)methyl)guanine	Hypoxanthine	C_12_H_18_N_6_O_4_	0.04	0.044
plp8_60_266_1595m_z	yes	Prenyl glucoside	Hemiterpenoid glycoside	C_11_H_20_O_6_	0.04	0.044
plp6_45_327_1195m_z		Ethyl 8-azido-5-methyl-6-oxo-4H-imidazo [1,5-a][1,4]benzodiazepine-3-carboxylate	Imidazo [1,5-a][1,4]benzodiazepines	C_15_H_14_N_6_O_3_	0.0007	0.004
plp3_52_347_2614m_z		Methyl-[10]-shogaol	Dimethoxybenzene	C_22_H_34_O_3_	0.002	0.006
plp6_70_344_1229n		Unknown		C_19_H_20_O_6_	0.003	0.006
slp0_28_976_8400m_z	yes	TG 60:8	Triacylglycerol (polyunsaturated)	C_63_H_106_O_6_	0.002	0.004
** *Forced vital capacity (FVC) at 10, 18, and/or 26 years* **						
plp7_23_745_6112m_z		Unknown	Unknown		0.005	0.01
plp1_36_334_2136n	yes	Many isomers possible	Diterpenoid (retinoid) or oxylipin	C_20_H_28_O_3_	0.02	0.02
plp1_41_282_1939n	yes	Unknown	Unknown	C_15_H_23_NO_3_	0.02	0.02
plp1_47_156_0786n	yes	Unknown	Unknown	C_8_H_12_O_3_	0.03	0.03
plp1_62_270_1695m_z	yes	Many isomers possible	Unknown	C_14_H_20_O_4_	0.001	0.003
plp1_99_534_3901m_z	yes	N-Decanoylsphingosine-1-phosphate (CerP(d18:1/10:0))	Sphingolipid	C_28_H_56_NO_6_P	0.008	0.01
plp5_20_364_1857m_z	yes	Phe-Pro-Thr (or isomer)	Tripeptide	C_18_H_25_N_3_O_5_	0.007	0.01
** *Forced expiratory volume in 1 s (FEV_1_) at 10 and/or 18 years ) (none)* **						
** *FEV1/FVC ratio at 10 and/or 18 years* **						
slp0_28_694_6473m_z		CE 20:2	Cholesterol ester	C_47_H_80_O_2_	0.01	0.01
** *Forced mid-expiratory flow FEF_25–75%_ at 10 and/or 18 years (none)* **						

^#^ For MNTs with significant interaction with time, the *p*-value column represents the *p*-value of the interaction term in the model. Without significant interaction, the *p*-value indicates the main effect in the model.

**Table 3 metabolites-13-00737-t003:** (**a**). Metabolites, nutrients, toxins (MNTs) conveying protective effects for allergic and respiratory outcomes in female F1-offspring. (**b**). Metabolites, nutrients, toxins (MNTs) conveying protective effects for allergic and respiratory outcomes in male F1-offspring.

**(a)**						
***Health Outcome* and Associated MNTs**	**Interaction with Time**	**Annotation**	**Compound Class**	**Chemical** **Formula**	***p*-Value ^#^**	**FDR Adjusted *p*-Value**
** *Asthma at 4, 10, 18, and/or 26 years (none)* **						
** *Skin prick test positivity at 4, 10, and/or 18 years* **						
plp0_85_380_3506m_z		Unknown	Unknown		0.002	0.01
plp0_90_468_3883m_z	yes	Unknown	Unknown		0.004	0.008
plp1_25_205_0968m_z		L-Tryptophan	Amino acid	C_11_H_12_N_2_O_2_	0.001	0.004
plp1_65_444_1957m_z		Met-Phe-Phe	Tripeptide	C_23_H_29_N_3_O_4_S	0.001	0.004
plp10_25_189_1597m_z	yes	N6,N6,N6-Trimethyl-L-lysine	Amino acid derivative, carnitine precursor	C_9_H_20_N_2_O_2_	0.008	0.01
plp6_13_119_0928m_z	yes	Unknown	Unknown		0.01	0.02
** *Immunoglobulin E levels at 10 and/or 18 years (none)* **						
** *Fractional exhaled nitric oxide (FeNO) at 18 and/or 26 years (none)* **						
** *Forced vital capacity (FVC) at 10, 18, and/or 26 years* **						
plp3_48_288_2066m_z		Unknown	Unknown	C_17_H_25_N_3_O	0.001	0.003
** *Forced expiratory volume in 1 s (FEV_1_) at 10,18, and/or 26 years* **						
slp0_28_397_3802m_z		Sitosterol fragment ion (reflects plant sterols)	Plant sterol	C_29_H_49_^+^	0.002	0.03
plp4_57_692_4474m_z		PS(14:1(9Z)/15:0)	Phosphatidyl serine, monounsaturated	C_35_H_66_NO_10_P	0.0009	0.008
plp6_91_180_0491m_z		Unknown	Unknown		0.002	0.013
** *FEV1/FVC ratio at 10, 18, and/or 26 years (none)* **						
** *Forced mid-expiratory flow FEF_25–75%_ at 10, 18, and/or 26 years* **						
plp0_86_340_3475m_z		Unknown	Unknown		0.0001	0.001
nlp16_65_861_5483m_z	yes	Unknown	Unknown		0.02	0.02
**(b)**						
***Health Outcome* and Associated MNTs**	**Interaction with Time**	**Annotation**	**Compound Class**	**Chemical Formula**	** *p* ** **-Value ^#^**	**FDR Adjusted *p*-Value**
** *Asthma at 4, 10, 18, and/or 26 years (none), Skin prick test positivity at 4, 10, and/or 18 years (none)* **						
** *Immunoglobulin E levels at 10 and/or 18 years (none), Fractional exhaled nitric oxide (FeNO) at 18 and/or 26 years (none)* **						
** *Forced vital capacity (FVC) at 10, 18, and/or 26 years* **						
plp1_59_201_1384m_z		Tetrahydrozoline	Imidazoline pharma- ceutical	C_13_H_16_N_2_	0.045	0.045
slp0_28_992_9463m_z	yes	TG 60:0	Triacylglycerol, fully saturated	C_63_H_122_O_6_	0.02	0.02
plp2_94_799_4893m_z	yes	Unknown	Unknown		0.00009	0.001
plp2_92_351_0460m_z	yes	Unknown	Unknown	C_6_H_16_N_4_O_9_P_2_	0.0002	0.001
plp2_94_279_0142m_z	yes	3,5-Dimethoxy-4-(sulfooxy)benzoic acid	Polyphenol (syringic acid) sulfate conjugate	C_9_H_10_O_8_S	0.0003	0.001
** *Forced expiratory volume in 1 s (FEV_1_) at 10, 18, and/or 26 years* **						
plp0_94_792_5631m_z		MGDG 36:6	Galactosylglycerol (plant) lipid	C_45_H_74_O_10_	0.01	0.01
plp2_94_137_0413n	yes	Hypoxanthine [^13^C_1_] isotopolog	Hypoxanthine	C_5_H_5_N_4_O	0.001	0.003
plp2_94_295_0654m_z	yes	Unknown	Unknown		0.002	0.004
plp2_94_799_4893m_z	yes	PG 40:9	Phosphatidylglycerol (polyunsaturated)	C_41_H_68_NO_11_P	0.0005	0.002
plp2_94_177_0647n	yes	Hypoxanthine, acetonitrile adduct	Hypoxanthine	C_5_H_5_N_4_O	0.002	0.004
plp2_95_143_0536m_z	yes	2,5-Dimethyl-3-(methylthio) furan	Aryl thioether	C_7_H_10_OS	0.0004	0.002
plp2_92_351_0460m_z	yes	Unknown	Unknown	C_6_H_16_N_4_O_9_P_2_	0.003	0.0045
plp2_94_279_0142m_z	yes	3,5-Dimethoxy-4-(sulfooxy)benzoic acid	Polyphenol (syringic acid), sulfate conjugate	C_9_H_10_O_8_S	0.004	0.005
** *FEV1/FVC ratio at 10, 18, and/or 26 years* **						
slp0_28_326_3818m_z		Didecyl dimethylammonium (DDAC) antiseptic	Antiseptic	C_22_H_48_N^+^	0.001	0.002
plp9_21_215_0557n		Glycero-3-phosphoethanolamine	Glycerophospho-ethanolamines	C_5_H_14_NO_6_P	0.0008	0.0009
plp9_21_260_0267m_z		Unknown	Unknown		0.0009	0.0009
** *Forced mid-expiratory flow FEF_25–75%_ at 10, 18, and/or 26 years* **						
slp0_28_326_3818m_z		Unknown	Unknown		0.002	0.002
plp4_08_257_0586m_z		7-Amino-4-hydroxy-2-naphthalenesulfonic acid	Dye precursor or breakdown product	C_10_H_9_NO_4_S	0.04	0.04

^#^ For MNTs with significant interaction with time, the *p*-value column represents the *p*-value of the interaction term in the model. Without significant interaction, the *p*-value indicates the main effect in the model.

**Table 4 metabolites-13-00737-t004:** Metabolites, nutrients, toxins (MNTs) whose risks were replicated in the F2-generation stratified by sex.

*Health Outcome* and Associated MNTs	Interaction with Time	Annotation	Compound Class	*p*-Value (F1)	FDR Adjusted *p*-Value (F1)	*p*-Value Replication (F2)
**Females F2**						
**Fractional exhaled nitric oxide (log10 of FeNO) at 6–7 years of age**						
plp1_60_369_2086m_z		Benzyl (2-ethylhexyl)phthalate	Phthalate plasticizer	0.002	0.01	0.0868
plp1_45_242_1558m_z		Desmethyldiphenhydramine	OTC Antihistamine/sedative metabolite	0.006	0.02	0.0050
**Males F2**						
**Asthma at 6–7 years of age**						
plp0_95_237_1019m_z		Glycosminine	Quinazoline alkaloid	0.004	0.012	0.0066
**Immunoglobulin E levels (log10 of IgE) at 6–7 years of age**						
plp5_70_588_3739m_z		3α-[(β-D-Glucopyranosyl)oxy]-7α,12α-dihydroxy-5 β-cholanic acid	Steroid metabolite; bile acid (cholic acid) glucoside	0.002	0.005	0.1308
plp5_90_231_1698m_z		Ile-Val; Val-Ile; Leu-Val; or Val-Leu	Dipeptide	0.002	0.005	0.0835
**Fractional exhaled nitric oxide (log10 of FeNO) at 6–7 years of age**						
plp1_57_497_2341m_z		Unknown	Unknown	0.002	0.006	0.1838
plp1_05_161_1074m_z	yes	Tryptamine	Tryptophan metabolite (indole)	0.0004	0.004	0.0046
plp8_60_266_1595m_z	yes	Prenyl glucoside	Hemiterpenoid glycoside	0.04	0.044	0.1478
slp0_28_976_8400m_z	yes	TG 60:8	Triacylglycerol (polyunsaturated)	0.002	0.004	0.0264

**Table 5 metabolites-13-00737-t005:** Metabolites, nutrients, toxins (MNTs) whose protective associations were replicated in the F2-generation stratified by sex.

*Health Outcome* and Associated MNTs	Interaction with Time	Annotation	Compound Class	*p*-Value (F1)	FDR Adjusted *p*-Value (F1)	*p*-Value Replication (F2)
**Females**						
** *Skin prick test positivity at 1, 3, and/or 6 years* **						
plp10_25_189_1597m_z	yes	N6,N6,N6-Trimethyl-L-lysine	Amino acid derivative	0.008	0.01	0.1699
plp6_13_119_0928m_z	yes	Unknown	Unknown	0.01	0.02	0.0732
** *Forced mid-expiratory flow FEF_25–75%_ at 6–7 years of age* **						
nlp16_65_861_5483m_z	yes	Unknown	Unknown	0.02	0.02	0.13
**Males**						
** *Forced vital capacity (FVC) at 6–7 years of age* **						
slp0_28_992_9463m_z	yes	TG 60:0	Saturated Triacylglycerol	0.02	0.02	0.0359
plp2_94_799_4893m_z	yes	PE 38:8	Phosphatidylethanolamine (polyunsaturated)	0.00009	0.001	0.0324
plp2_92_351_0460m_z	yes	Unknown	Unknown	0.0002	0.001	0.0871
plp2_94_279_0142m_z	yes	3,5-Dimethoxy-4-(sulfooxy)benzoic acid	Polyphenol sulfate	0.0003	0.001	0.0478
** *Forced expiratory volume in 1 s (FEV_1_) at 6–7 years of age* **						
plp2_94_137_0413n	yes	Hypoxanthine [^13^C_1_] isotopolog	Hypoxanthine	0.001	0.003	0.0111
plp2_94_295_0654m_z	yes	Unknown	Unknown	0.002	0.004	0.0156
plp2_94_799_4893m_z	yes	PE 38:8	Phosphatidylethanolamine (polyunsaturated)	0.0005	0.002	0.0581
plp2_94_177_0647n	yes	Hypoxanthine acetonitrile adduct	Hypoxanthine	0.002	0.004	0.1067
plp2_95_143_0536m_z	yes	2,5-Dimethyl-3-(methylthio)furan	Aryl thioether	0.0004	0.002	0.0398
plp2_92_351_0460m_z	yes	Unknown	Unknown	0.003	0.005	0.1575
** *Forced mid-expiratory flow FEF_25–75%_ at 6–7 years of age* **						
plp4_08_257_0586m_z		7-Amino-4-hydroxy-2-naphthalenesulfonic acid	Dye precursor or breakdown product	0.04	0.04	0.0145

## Data Availability

Metabolomics data have been deposited to the EMBL-EBI MetaboLights database [77] with the identifier MTBLS6941. The complete dataset can be accessed here: https://www.ebi.ac.uk/metabolights/MTBLS6941 (accessed on 7 June 2023).

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
