# Peer review of "Association of Metabolites, Nutrients, and Toxins in Maternal and Cord Serum with Asthma, IgE, SPT, FeNO, and Lung Function in Offspring"

_metabolites, 2023, doi:10.3390/metabo13060737_

Round 1

Reviewer 1 Report

In this cohort based study, the authors investigated the correlation between grandmaternal and paternal serum nutrients, toxins and metabolites in offspring in paternal children and grandchildren. Nine detected substances were found to be correlated with a higher risk for respiratory allergic outcomes and twelve were found to be protective.

This is a very interesting study, as it uses a kind of shotgun-method to identify risk-marker associations and is not driven by a mechanistic hypothesis. It may thus report unexpected new associations. The English text is very well written.

The term "MNT" appears to be an attempt to "encompass" a very large set of very different substances. We understand that the term is used give a single name to a large heterogeneous group of compounds that are not always functionally or structurally related. However, it should be made clear that these substances do not have a common feature (except serum abundance), as the common abbreviation may otherwise suggest.

A (probably better) suitable citation than [25] for ComBat may be:

Johnson WE, Li C, Rabinovic A. Adjusting batch effects in microarray expression data using empirical Bayes methods. Biostatistics. 2007;8:118–27. pmid:16632515

Line 174: "However, the principal components of these stable lipids were not significantly different across batches." How do we know, if the model is sensitive to lipid oxidation? The model (principal component analysis) should be described more extensively. An experimental (or theoretical) prove for lipid oxidation sensitivity within the model should be provided.

It is not necessary to show all study-internal identifiers like "slp0_28_326_3818m_z". They make reading harder without providing additional information. Common substance names should be used if possible. Also the heading of table 4 seems to be misleading, as "associated outcome" is used for the column with identifiers. Showing the identifiers may not be necessary here as well, as the reader cannot make use of them..

The discussion of detected associations needs to include more available literature:

Line 518-527: Quinazalones are also used for treatment of asthma (bronchodilators) and may thus also be affected by indication bias.

Line 528-533: Prenyl glucosides are also known anti-asthmatic substances, for example used in form of natural asthma treatment (f.e. in Capparis spinosa) https://core.ac.uk/download/pdf/109947457.pdf

Line 548-464: Tryptophan metabolism has long been suggested to be relevant in the pathophysiology of allergic disorders, including asthma. Literature should be provided.

Line 565-576: Elevated serum triglycerides were associated with the presence of asthma in patients with obesity in Allergy Asthma Proc, 2021 May 1;42(3):e71-e76., doi: 10.2500/aap.2021.42.210020. Anyway, given associations in dependence of BMI would be also very interesting in this context.

Serum levels of hypoxanthine were markedly higher in asthmatic subjects compared with those in COPD or healthy subjects (doi: 10.3967/bes2019.085).

Anyway, appearant accordance with existent reports clearly indicates that the correlation method generally works and provides relevant information. The provided set of completely new correlations needs confirmation, but is highly interesting.

Minor points:

Reference error in line 366, line 427)

Author Response

Note to the reviewer: We would like to thank the editor and reviewer for their interest in our manuscript and the insightful, careful, and constructive comments and suggestions that improved our manuscript.

  1. A (probably better) suitable citation than [25] for ComBat may be: Johnson WE, Li C, Rabinovic A. Adjusting batch effects in microarray expression datausing empirical Bayes methods. Biostatistics. 2007;8:118–27. pmid:16632515
    Answer: As suggested by the reviewer, we included the methodological paper of ComBat (Johnson et al. 2007) in the revised manuscript.

  1. Line 174: "However, the principal components of these stable lipids were not significantly different across batches." How do we know, if the model is sensitive to lipid oxidation? The model (principal component analysis) should be described more extensively. An experimental (or theoretical) prove for lipid oxidation sensitivity within the model should be provided.

Answer: As described, principal component analysis was performed using only the signals for the stable lipids to assess whether drift in instrument performance was significant across the 35 batches of sera analyzed over the period of March 2018 to July 2019.  As stated in the manuscript, this PCA analysis did not include signals from polyunsaturated lipids expected to be more reactive toward oxidation, nor did it include oxidized forms of lipids that were detected. A separate PCA analysis was performed that included all MNTs detected in the negative-ion lipid analysis, and the PCA scores plot showed clustering of F0 sera separate from F2 sera, driven primarily by the greater extent of lipid oxidation in F0 following the much longer storage period (~ 30 years).

  1. It is not necessary to show all study-internal identifiers like "slp0_28_326_3818m_z". They make reading harder without providing additional information. Common substance names should be used if possible.

Answer: While we appreciate the reviewer’s suggestion that readability might be improved if the identifiers were replaced with common names, in numerous cases, the common names of compounds are quite lengthy and likely to be less readable (a few have more than 70 characters in their names). In addition, we have concerns that “common substance names” may not confer much useful information, particularly for non-chemists.  The current nomenclature does serve to indicate which kind of analysis detected the reported MNTs.  The current nomenclature may also help to identify several yet unknown substances. However, we doubt that substituting names would generally improve readability without sacrificing some of the information we wish to convey about the analysis needed to detect them.  When re-numbering compounds in the tables (e.g., reporting this in the text as Compound SLP 1, while retaining and including the compounds’ numbers with the identifiers in the manuscript text), we doubt whether this will improve readability of the manuscript.

  1. Also the heading of table 4 seems to be misleading, as "associated outcome" is used for the column with identifiers. Showing the identifiers may not be necessary here as well, as the reader cannot make use of them.

Answer:  We changed the headings of column 1 used in Table 2(a) to Table 5 to.

Health outcome and associated MNTs”.

  1. Line 518-527: Quinazalines are also used for treatment of asthma (bronchodilators) and may thus also be affected by indication bias.

Answer: We recognize that quinazoline have been reported as PDE4 inhibitors for asthma treatment (Elansary et al. Med Chem Res (2012) 21:3327–3335) and that some quinazoline derivatives have been marketed as anti-inflammatory drugs (e.g., NSC127213), but our analysis did not detect these compounds or any anticipated metabolites.

However, we agree and changed the discussion. We deleted the following sentence:

“Further studies are needed to establish the occurrence of glycosminine in commonly consumed citrus fruits and investigate whether the quinazoline alkaloid is a risk factor or can be explained by a spurious association.”

We added the following sentences:

“However, some quinazoline derivates also provide bronchodilator activity [1]. Hence, the reported association could also be due to an indication bias. In this case mothers with asthmatic symptoms used bronchodilators during pregnancy and their offspring were more likely to develop asthma. This, then results in a spurious association between quinazoline and asthma in offspring.”

In addition, we deleted “a quinazoline alkaloid,” from the Abstract.

  1. Line 528-533: Prenyl glucosides are also known anti-asthmatic substances, for example used in form of natural asthma treatment (f.e. in Capparis spinosa) https://core.ac.uk/download/pdf/109947457.pdf

Answer: Beneficial effects on coughs and asthma have been reported for Capparis spinosa [2, 3] which; however, capper contains multiple phytochemicals (alkaloids, flavonoids, glucosinolates, phenolic acids, terpenoids and others). The potential anti-asthmatic effect has not been linked to one phytochemical substance such as prenyl glucosides. In fact, our findings suggest increased levels of prenyl glucoside were associated with increases in offspring FeNO, a finding at odds with the view that they are anti-asthmatic. Prenyl glucoside has been observed in other foods besides citrus and capers (e.g., in fennel; Kitajima et al., (1998) Chem. Pharm. Bull. 46: 1643-6) but has seldom been described in the literature.

  1. Line 548-464: Tryptophan metabolism has long been suggested to be relevant in the pathophysiology of allergic disorders, including asthma. Literature should be provided.

Answer:  The reviewer is correct that numerous studies have documented associations of various tryptophan metabolites with asthma and immune system functions.  This section of our manuscript references six publications (references 55-60) that address various aspects, but our study differs in that the metabolites were measured in maternal sera or cord blood collected at birth and lung function assessments were made later in life. Hence – as stated in the discussion - our investigation is not biased by reverse causation (e.g., asthma à tryptophan metabolites).

  1. Line 565-576: Elevated serum triglycerides were associated with the presence of asthma in patients with obesity in Allergy Asthma Proc, 2021 May 1;42(3):e71-e76., doi:10.2500/aap.2021.42.210020. Anyway, given associations in dependence of BMI would be also very interesting in this context.

Answer: We agree that an analysis of obesity (BMI) regarding triglycerides would be of interest. However, first, this would require a different framework and would blast the length of this article. Second, we were not measuring serum triglycerides in general but very specific lipid metabolites. Third, the referenced article by van Zelst (doi:10.2500/aap.2021.42.210020) investigates the role of triglycerides in the presence of asthma using a cross-sectional analysis. We have a different time order analyzing lipids in maternal serum before birth (F1) or cord serum after delivery (F2) and the development of asthma in offspring later in life. We discussed whether to take maternal BMI before birth into account or the BMI when asthma developed in offspring (or both). Anyway, both links and their combination would be of interest but require additional conceptual approaches and would result in an additional publication.

  1. Serum levels of hypoxanthine were markedly higher in asthmatic subjects compared with those in COPD or healthy subjects (doi: 10.3967/bes2019.085).

Answer: The study by Liang et al. 2019 “Metabolomic Profiling Differences among Asthma, COPD, and Healthy Subjects - A LC-MS-based Metabolomic Analysis” investigated diseased adults and compared them with healthy subjects. Whereas this study can be affected by reverse causation (asthma or COPD result in changes of hypoxanthine), our study has a clear time order from birth to later in childhood or adulthood, which shows a protective association. Hence, we added to the discussion:

“Liang et al. [69] in a cross-sectional analysis compared diseased (asthma and COPD) and healthy adults. Contrary to our findings, the authors suggested that hypoxanthin constitutes a risk. However, other than this cross-sectional comparison, our study has a clear time order from birth to later in childhood or adulthood and assesses longitudinal associations.”

  1. Reference error in line 366, line 427

Line 366: “Although serum levels of thiamine did not show significance in this study, we assumed > 100-fold lower mean levels measured in F0 sera relative to F2 sera can be attributed to thiamine decomposing during 30+ years of serum storage and not presenting a useful compound for associations in F0.”

We would need to add two references supporting our statement.

  • Wagner-Golbs A, Neuber S, Kamlage B, Christiansen N, Bethan B, Rennefahrt U, Schatz P, Lind L. Effects of Long-Term Storage at -80 °C on the Human Plasma Metabolome. Metabolites. 2019 May 17;9(5):99. doi: 10.3390/metabo9050099. PMID: 31108909; PMCID: PMC6572224.
  • Haid M, Muschet C, Wahl S, Römisch-Margl W, Prehn C, Möller G, Adamski J. Long-Term Stability of Human Plasma Metabolites during Storage at -80 °C. J Proteome Res. 2018 Jan 5;17(1):203-211. doi: 10.1021/acs.jproteome.7b00518. Epub 2017 Nov 28. PMID: 29064256.

Line 427: “Among MNTs that were risk factors in girls, plp1_60_369_2086m_z, benzyl (2-ethylhexyl) phthalate, a phthalate plasticizer, increased FeNO in the F1 generation (Figure S2.E) and was replicated in F2 (Table 4).”

            We corrected the reference to the figures/table.

Reviewer 2 Report

The study was designed to explore the risk of respiratory and allergic health outcomes in offspring and the association with serum levels of MNTs at the end of pregnancy. The cohort was recruited between January 1, 1989, and February 1990. Each offspring was followed at various time points until the age of 26. The manuscript was well written.  One of the major factors that the authors should consider is the seasonality of respiratory/allergic disorders. Therefore the timing of the blood collection is critical and should be incorporated in the data analysis.

Author Response

Note to the reviewer: We would like to thank the editor and reviewer for their interest in our manuscript and the insightful, careful, and constructive comments and suggestions that improved our manuscript.

One of the major factors that the authors should consider is the seasonality of respiratory/allergic disorders. Therefore, the timing of the blood collection is critical and should be incorporated in the data analysis.

Answer: The seasonality of respiratory/allergic disorders did not play a role, which is not surprising since the appointment for the clinical assessments with was moved when the participants suffered from an infection (see Method section).

Nevertheless, the timing of the blood collection constitutes an important critique. We checked the timing of the blood collection and found multiple statistically significant associations with MNTs for the F1 generation, but surprisingly none were replicated in the F2 generation. To understand this, we need to consider that the F1 generation were all born in in about one year (from January 1989 to February 1990), whereas the F2 generation was born over 10 years (between 2010-2019). Hence, the effect of seasonality in the F2 generation was potentially diluted by an association with year of birth. Hence, we additionally took year of birth into account but could not replicate the season findings of F1. This setting was not improved when we looked a month of birth, since the season varied with year. Based on these findings, we did not follow potential biases which could theoretically have been related to season on birth.

Conceptually, on purpose, we did not include season of maternal blood collection before birth of the child in the F0-generation or season of cord blood collection in the F2-geneatrion. The reason is that season of birth can turn the MNTs measured at birth into mediating variables:  season of birth  à MNT à allergic or respiratory outcomes. If we statistically control for season of birth, the associative arm of MNT à allergic or respiratory outcomes would disappear, and season of birth would become the driving exposure. There are multiple studies showing a link between season of birth and allergic or respiratory outcomes, but it is unknown which chemical substances may produce such a link. Here we show the chemical substances that produce that are related to allergic or respiratory outcomes.

Reviewer 3 Report

The manuscript “Association of metabolites, nutrients, and toxins in maternal and cord serum with asthma, IgE, SPT, FeNO, and lung function in offspring” presents an interesting approach to investigate the association between metabolites, nutrients, and toxins, (shortened to MNTs) and asthma. The experiments are well planned, the analysis well described and carefully performed and confounding factors taken into consideration. Of particular interest is the correlation between certain MNTs and lung function and the protective effect of a polyunsaturated phosphatidylglycerol. The text is well written and easy to follow. Taken together, the manuscript contributes to the understanding of asthma and allergy development. One drawback of the experiments is that the results give no causative insight into the diseases, but may contribute to such understanding in the future. There are no major concerns with the manuscript.

Author Response

Note to the reviewer: We would like to thank the editor and reviewer for their interest in our manuscript and the insightful, careful, and constructive comments and suggestions that improved our manuscript.

One drawback of the experiments is that the results give no causative insight into the diseases, but may contribute to such understanding in the future. There are no major concerns with the manuscript.

No response was necessary.

Round 2

Reviewer 2 Report

I have no further comments.